# Integrated Transcriptome and Metabolome Analysis Reveals the Molecular Mechanism of Rust Resistance in Resistant (Youkang) and Susceptive (Tengjiao) *Zanthoxylum armatum* Cultivars

**DOI:** 10.3390/ijms241914761

**Published:** 2023-09-29

**Authors:** Shan Han, Xiu Xu, Huan Yuan, Shujiang Li, Tiantian Lin, Yinggao Liu, Shuying Li, Tianhui Zhu

**Affiliations:** 1College of Forestry, Sichuan Agricultural University, Chengdu 611130, China; 13722@sicau.edu.cn (S.H.); xuxiu@stu.sicau.edu.cn (X.X.); yuanhuan@stu.sicau.edu.cn (H.Y.); 14087@sicau.edu.cn (S.L.); tlin@sicau.edu.cn (T.L.); 11468@sicau.edu.cn (Y.L.); 72111@sicau.edu.cn (S.L.); 2Key Laboratory of Forest Protection of Sichuan Education Department, Sichuan Agricultural University, Chengdu 611130, China; 3Key Laboratory of National Forestry & Grassland Administration on Forest Resources Conservation and Ecological Safety in the Upper Reaches of the Yangtze River, Sichuan Agricultural University, Chengdu 611130, China

**Keywords:** *Zanthoxylum armatum*, *Coleosporium zanthoxyli*, transcriptome, metabolome, disease resistance, phenylpropanoid metabolism

## Abstract

Chinese pepper rust is a live parasitic fungal disease caused by *Coleosporium zanthoxyli*, which seriously affects the cultivation and industrial development of *Z. armatum*. Cultivating and planting resistant cultivars is considered the most economical and environmentally friendly strategy to control this disease. Therefore, the mining of excellent genes for rust resistance and the analysis of the mechanism of rust resistance are the key strategies to achieve the targeted breeding of rust resistance. However, there is no relevant report on pepper rust resistance at present. The aim of the present study was to further explore the resistance mechanism of pepper by screening the rust-resistant germplasm resources in the early stage. Combined with the analysis of plant pathology, transcriptomics, and metabolomics, we found that compared with susceptible cultivar TJ, resistant cultivar YK had 2752 differentially expressed genes (DEGs, 1253 up-, and 1499 downregulated) and 321 differentially accumulated metabolites (DAMs, 133 up- and 188 down-accumulated) after pathogen infection. And the genes and metabolites related to phenylpropanoid metabolism were highly enriched in resistant varieties, which indicated that phenylpropanoid metabolism might mediate the resistance of *Z. armatum*. This finding was further confirmed by a real-time quantitative polymerase chain reaction analysis, which revealed that the expression levels of core genes involved in phenylpropane metabolism in disease-resistant varieties were high. In addition, the difference in flavonoid and MeJA contents in the leaves between resistant and susceptible varieties further supported the conclusion that the flavonoid pathway and methyl jasmonate may be involved in the formation of Chinese pepper resistance. Our research results not only help to better understand the resistance mechanism of *Z. armatum* rust but also contribute to the breeding and utilization of resistant varieties.

## 1. Introduction

Chinese pepper, which belongs to the Rutaceae family, is a type of fragrant deciduous thorny shrub that is widely loved by Asians. As early as 2000 years ago, Chinese people began to plant and eat Chinese pepper [1]. Today, two pepper species are widely cultivated in China, namely, *Zanthoxylum bungeanum* Maxim (red pepper) and *Zanthoxylum armatum* DC (green pepper), with a planting area of more than 1.6 million ha and an economic value of more than USD 2 billion [2]. However, as the planting area of Chinese pepper increases, it is affected by various plant diseases during its growth period. Rust is a catastrophic disease in the production of Chinese pepper. In severe cases, the incidence rate can reach 100% and the defoliation rate can exceed 90%, affecting the differentiation of flower buds in the next year, causing an annual reduction of more than 20% in the production of Chinese pepper [3,4], greatly limiting the development of the Chinese pepper industry.

Rust, which is one of the top 10 fungal plant pathogens globally, has a wide host range and is highly destructive [5,6]. At present, the main strategy to control rust is to use chemical pesticides and to cultivate resistant varieties [7,8]. Chemical pesticides have their own advantages, but abuse of these pesticides can result in pesticide residues, pathogen resistance, and the resurgence of pests [9,10]. Sustainable management strategies for harmful organisms have become the main guiding direction for plant disease control, with an increasing emphasis on human health and environmental protection. The breeding and application of disease-resistant varieties can better solve the problem of plant disease prevention and control. However, there are currently no systematic research reports on the resistance of *Z. armatum* to rust, which has led to a lack of knowledge about rust resistance in *Z. armatum* varieties. Therefore, we decided to utilize the resistance resources of *Z. armatum* and to screen and identify resistance genes related to *Z. armatum* rust resistance, which is of great significance for the prevention and control of this disease.

As a biological stress factor, pathogenic bacteria can stimulate plant stress responses, activate defense mechanisms, and prevent diseases. The plant response to pathogen infection is related to large-scale changes in gene expression and metabolism [11], and transcriptome analysis provides a useful tool to identify genes that may contribute to plant resistance [12,13]. In addition, plants have complex metabolic regulatory networks that can play an important role in plant growth under pathogen infection conditions. At present, many studies have confirmed that the relationship between gene expression and metabolite biosynthesis can be widely analyzed by combining transcriptomics and metabolomics [11]. The combination of transcriptomics and metabolomics provides a powerful method for the in-depth and comprehensive research of plant defense response mechanisms against pathogen infections at the molecular and cellular levels [14,15]. For example, Jedrzej conducted a multiomics conjoint analysis to identify genomic loci related to the levels of hundreds of transcripts and metabolites and analyzed the disease resistance associated with the fruit ripening process of wild tomato [16]. Li et al. [17] combined transcriptomics and metabolomics analyses with the antifungal activity of flavonoids extracted from the stems of *Z. bungeanum* and found that flavonoid metabolism was regulating the resistance of *Z. bungeanum*. Furthermore, the use of plant secondary metabolites has gained popularity in the control of plant pathogen attacks [18]. Sade et al. [12] found that phenylpropanoid and tryptophan metabolism were enriched to varying degrees after infection with tomato yellow leaf curl virus. Therefore, a correlation analysis of annotation results of differentially accumulated metabolites (DAMs) and differentially expressed genes (DEGs) to metabolic pathways can better explain the transcriptional regulation mechanism of the plant response to pathogen infections.

*Zanthoxylum armatum* is an important seasoning, spice, and woody oil species in China. However, the high incidence rate of Chinese pepper rust has become the main bottleneck restricting the production of Chinese pepper. The selection of resistant varieties is one of the most important methods for controlling rust, but there are currently no relevant reports on the mechanism of resistance to Chinese pepper rust. In the current study, we first evaluated the occurrence of leaf rust in two pepper varieties and selected the resistant variety ‘Youkang’ (YK) and the susceptible variety ‘Tengjiao’ (TJ). Second, to clarify the resistance mechanism of *Z. armatum* to rust, transcriptomic and metabolomic analyses were conducted to analyze the differences in gene expression and metabolite biosynthesis between resistant and susceptible varieties. In addition, the correlation between gene expression and metabolite biosynthesis was analyzed to better understand the resistance mechanism of *Z. armatum* to rust. This study provides information on the response of *Z. armatum* to rust infection, revealing an important biological pathway for Chinese pepper’s resistance to rust. This result may provide new insights into the pathogenesis of Chinese pepper disease and greatly promote the selection of resistant varieties.

## 2. Results

### 2.1. Analysis of Rust Resistance of ‘Tengjiao’ and ‘Youkang’ Chinese Pepper Leaves

In our previous experiments, we conducted a four-year investigation of the disease severity in a segregated Chinese pepper population containing 200 individuals. Under field conditions, it was observed that in YK, leaf rust caused small disease spots, the incidence rate was less than 30%, and the disease index was 12.67. By contrast, TJ leaves showed a large area of rust infection, with an incidence rate of more than 80% and a disease index of 52.24 (Figure 1A). To eliminate the possible impact of environmental factors, we further selected healthy leaves of TJ and YK, and detached leaves were inoculated with *Coleosporium zanthoxyli*. At 16 days post-inoculation (dpi), the lesions on the TJ leaves began to spread and further increased significantly. The lesions on the YK leaves spread less (Figure 1B). In addition, the infection rate of *C. zanthoxyli* on the TJ leaves was higher than that on the YK leaves (Figure 1C). These results indicate that the YK leaves were resistant to *C. zanthoxyli*.

### 2.2. Transcriptome Data Analysis

#### 2.2.1. RNA Quality Determination and Sequencing Results

An RNA-Seq analysis was performed on each variety (YK and TJ) at 12 dpi in three independent replicates to demonstrate the potential resistance mechanism of *Z. armatum* to rust. A total of 12 libraries (YK-CK (A), YK-T (B), TJ-CK (C), and TJ-T (D), each with three biological replicates) were constructed, resulting in 5.7–6.3 Gb of clean reads with Q20 values greater than 97.22% and Q30 values greater than 92.44% (Table 1). After assembling clean reads, a total of 89,348 unigenes were obtained, with a maximum length of 16,841 bp, an average length of 1036 bp, an N50 length of 1461 bp, and an average GC% ratio of 44.34%. An analysis of the length distribution of unigenes showed that the unigenes with a length of less than 600 bp account for about 72% of the total number. A principal component analysis (PCA) was performed on the gene expression profiles of the 12 samples (Figure 2A), indicating that significant differences between the infected and control plants accounted for PC1 (38.03%), while differences between the pepper varieties mainly accounted for PC2 (18.79%). A correlation analysis was conducted on the gene expression levels between the samples (Figure 2B), and the results show a high consistency among the same groups of samples. Overall, the quality of the sequencing data was high, and the data met the requirements for further analysis.

#### 2.2.2. Transcriptome-Wide Identification of DEGs and Functional Analysis

The DEGs analysis results of the infected and control combinations of resistant and susceptible Chinese pepper are shown in Figure 3. A Venn diagram analysis showed 57,452, 56,742, 54,881, 55,893 co-expression DEGs, respectively, from the pairwise comparisons of four groups (Figure 3A). According to the expression levels of DEGs in the TJ-CK, TJ-T, YK-CK, and YK-T leaves, all DEGs were clustered into nine groups (Figure 3B). Based on the fragments per kb per million reads (FPKMs) values, we identified 2041, 3068, 10,807, and 2752 DEGs in the four comparisons (YK-CK vs. YK-T, TJ-CK vs. TJ-T, TJ-CK vs. YK-CK, and TJ-T vs. YK-T, respectively) through a transcriptome differential expression analysis (DESeq), in which 1349, 2153, 6432, and 1253 DEGs were upregulated and 692, 915, 4375, and 1499 DEGs were downregulated, respectively (Figure 2B and Appendix A). This result indicates that after rust infection, the gene expression levels in the leaves of resistant and susceptible varieties of *Z. armatum* have significantly changed. The heatmap in Figure 3C shows a summary of the expression levels of all identified DEGs.

#### 2.2.3. GO and KEGG Pathway Enrichment Analysis of DEGs in ‘Tengjiao’ and ‘Youkang’ Infected with *C. zanthoxyli*

Upon *C. zanthoxyli* infection, Chinese pepper has many DEGs. A Gene Ontology (GO) annotation analysis of the transcriptome of the *Z. armatum* leaves revealed the 15 most abundant GO terms in all the DEGs in different comparisons, including the molecular function, cellular component, and biological process categories. There are 25 annotations related to biological processes, among which “metabolic processes”, “cellular processes”, and “single biological processes” are the most abundant. There are 24 annotations related to cellular components, among which “membrane”, “cell” and “cell part” are the most abundant. There are 19 annotations related to molecular functions, among which “binding” and “catalytic activities” are the most abundant (Figure 4A). Moreover, the DEGs related to *C. zanthoxyli* infection in YK are mainly enriched in terms of “membrane part”, “inherent component of the membrane”, “overall component of the membrane”, “membrane”, “redox process”, and “oxidoreductase activity”. The DEGs related to disease resistance and rust resistance of Chinese pepper are mainly enriched in “membrane part”, “internal components of the membrane”, “overall components of the membrane”, “membrane”, “redox process” and “oxidoreductase activity” (Figure 4A).

A Kyoto Encyclopedia of Genes and Genomes (KEGG) enrichment analysis showed that 1930 DEGs were annotated to 121 metabolic pathways, among which 367 DEGs were significantly enriched in “photosynthesis”, “ribosomes”, “carbon metabolism”, “glyoxylate and dicarboxylate metabolism”, “carbon fixation in photosynthetic organisms”, “ABC transporters”, “flavonoid biosynthesis”, “alanine”, “aspartate and glutamate metabolism”, and “phenylalanine metabolism.” The 20 most significant pathways were selected to draw scatter plots (Figure 4B–E). Of these, 88 DEGs were annotated to the carbon metabolism pathway and 113 DEGs were annotated to the ribosome. Our KEGG pathway enrichment analysis results suggest that carbohydrate metabolism and ribosome-related genes play essential roles in the resistance to *Z. armatum* rust.

The KEGG pathway enrichment analysis of the DEGs showed that there were six common pathways between the comparisons of TJ-CK vs. TJ-T and YK-CK vs. YK-T, among which “phenylpropanoid biosynthesis” and “Plant hormone signal transduction” were the most significantly enriched pathways (Figure 4B,C). In addition, the “Biosynthesis of secondary metabolites” and “Flavonoid biosynthesis” pathways were highly enriched in the comparisons of TJ-CK vs. YK-CK and TJ-T vs. YK-T, which indicated that “phenylpropanoid biosynthesis” might be the pathway that is the most strongly related to the resistance of *C. zanthoxyli* infection (Figure 4C,E).

To analyze the flavone biosynthesis in the *Z. armatum* leaves in response to *C. zanthoxyli* infection, we annotated the genes encoding the enzymes HCT, 5-O-(4-coumaroyl)-D-quinate 3′-monooxygenase (C3′H), chalcone isomerase (CHI), flavonoid 3′-monooxygenase (F3′M), flavonoid 3′,5′-hydroxylase (F3′5′H), naringenin 3-dioxygenase (F3H), and flavonol synthase (FLS) (Figure 4). Most of them were significantly upregulated in YK in response to *C. zanthoxyli* infection, which may result in a boost in enzymatic activities and an increase in the accumulation of flavones.

#### 2.2.4. Verification by qRT-PCR

To verify the accuracy and repeatability of our transcriptome analysis, a qRT-PCR analysis was performed on a group of DEGs related to phenylpropane metabolism and jasmonic acid metabolism. Ten genes related to phenylpropane metabolism and jasmonic acid metabolism were selected from all the candidate DEGs for the qRT-PCR analysis. The expression profiles of these 10 DEGs as determined by qRT-PCR were consistent with the corresponding FPKM values obtained from the RNA-Seq analysis. These results indicate that the expression data obtained through RNA-Seq in this study are reliable (Figure 5).

### 2.3. Metabolome Data Analysis

#### 2.3.1. Data Quality Control

A widely targeted metabolome analysis was performed on samples taken from susceptible (TJ) and resistant (YK) leaf samples following *C. zanthoxyli* infection at 12 dpi (Figure 6). A total of 467 metabolites were obtained in all the samples, as presented in the heatmap, which shows a distinct hierarchical clustering of the samples by variety (Figure 4A). A heatmap of the DAMs confirmed the significant differences in the metabolomes of the TJ and YK leaf samples following *C. zanthoxyli* infection (Figure 6A). Based on the quantitative analyses of all the detected metabolites and the fold change (FC) threshold, a total of 182 DAMs were obtained in the comparison of TJ-T vs. YK-T, of which 112 metabolites were downregulated and 70 metabolites were upregulated in TJ-T. The *R*^2^ value between the quality control (QC) samples was close to 1 (Figure 6B), indicating that the whole testing process is reliable and that the quality of the data is excellent. 

#### 2.3.2. Metabolite Pathways and Classification Notes

According to the Human Metabolome Database (HMDB) (Figure 7A), 180 metabolites are lipids and lipid-like molecules, 147 metabolites are phenylpropanoids and polyketones, and 97 metabolites are organic heterocyclic compounds. The metabolites identified by the secondary profiles were subjected to a KEGG pathway analysis (Figure 7B). Most metabolites were annotated to amino acid metabolism, the biosynthesis of other secondary metabolites, and carbohydrate metabolism. LIPIDMAPS annotation was performed on the identified metabolites, and the results show that the metabolites were mainly enriched in flavonoids, fatty acids and conjugates, and isoprenoids (Figure 7C).

#### 2.3.3. DAM Screening and KEGG Pathway Annotation

The metabolite annotation results are shown in Figure 8. Metabolite screening was conducted in four groups, and a total of 1368 DAMs were screened (49 upregulated and 877 downregulated) (Figure 8A). The DAMs were annotated to 22 KEGG pathways. Based on these results, bubble maps of the top 20 enriched pathways were drawn, as shown in Figure 6B–E. The KEGG pathway analysis showed that DAMs were mainly enriched in the following seven pathways: phenylalanine metabolism, flavonoid biosynthesis, flavone and flavonol biosynthesis, nicotinate and nicotinamide metabolism, purine metabolism, plant hormone signal transduction, and glycine, serine, and threonine metabolism. Among them, the pentose phosphate pathway, carbon fixation in photosynthetic organisms, and flavone and flavonol biosynthesis were the most enriched.

The DAMs in the *Z. armatum* leaves in response to *C. zanthoxyli* infection were mainly classified into 10 categories, including flavonoids (45), phenolic acids (29), alkaloids (25), lignans and coumarins (24), lipids (22), organic acids (6), nucleotides and derivatives (6), amino acids and derivatives (6), terpenoids (4), and others (15) (Figure 8F). Among them, the accumulation of metabolites involved in the biosynthesis pathways of “phenylpropanoids and polyketides” and “lipids and lipid-like molecules” is the highest (Figure 8F), which is consistent with the RNA-Seq data.

#### 2.3.4. Metabolomic Analysis of TJ and YK Infected by *C. zanthoxyli*

The KEGG analysis also demonstrated that compared with TJ-T, flavonoid metabolism in YK-T was significantly enriched through three pathways: “flavonoid biosynthesis”, “flavonoid and flavonol biosynthesis”, and “isoflavone biosynthesis” (Figure 9A). Compared with TJ-T, DAMs in YK-T are mainly enriched in the biosynthesis of secondary metabolites in plants, such as phenylpropanes, flavonoids, and flavonoids (Figure 9A). Among the top 10 upregulated DAMs, two are flavonoid compounds (neohesperidin and hesperetin 5-O-glucoside) (Figure 9B). This result indicates that rust infection could induce a higher accumulation of flavonoids in the resistant variety YK. In summary, it could be speculated that phenylpropane metabolism contributes to the resistance of *Z. armatum* to leaf rust.

In addition, in this study, the upregulated flavonoids included flavanones (7), flavonoids and flavonols (6), chalcones and dihydrochalcones (3), one isoflavonoid (1), and one anthocyanidin (1). The clear enrichment of the flavonoid DAMs in the pepper leaves infected with rust fungus suggested that upregulated flavonoid compounds may be positively correlated with YK resistance. Moreover, the KEGG enrichment analysis also showed significant enrichment in the “phenylpropane metabolic synthesis” and “flavonoid and flavonol biosynthesis” pathways (Figure 8E). Therefore, we speculate that phenylpropane and flavonoid metabolism contributes to the resistance of *Z. armatum* to rust.

#### 2.3.5. Transcriptome and Metabolome Association Analysis

To further exploit the relationship between the DEGs and DAMs in the *Z. armatum* leaves in response to *C. zanthoxyli* infection, a co-expression network analysis of the transcriptome and metabolome was conducted for the comparison of TJ-T vs. YK-T. The results of the association analysis of TJ-T vs. YK-T showed that the DEGs and DAMs enriched in “flavonoid biosynthesis”, “glycerophospholipid metabolism”, “phenylpropane biosynthesis”, and “tryptophan metabolism” showed the same expression pattern, but there was a significant difference (*p* < 0.05). Among them, only the DEGs and DAMs enriched in the “phenylpropanoid biosynthesis” and “flavonoid biosynthesis” pathways showed extremely significant differences (*p* < 0.01), further confirming the conclusion that the resistance of YK against *C. zanthoxyli* infection was mainly positively regulated by phenylpropanoid biosynthesis and flavonoid biosynthesis. The most abundant KEGG pathway is “plant hormone and signal transduction”, which is related to one DAM and 28 DEGs (Figure 10A,B).

For the transcriptome analysis, the DEGs involved in the “plant hormone signal transduction” pathway was significantly differentially expressed in the comparison of TJ-T vs. YK-T (Figure 10C,D). For the metabolomic analysis, a significantly upregulated plant hormone, namely, methyl jasmonate (MeJA), was detected in the comparison of TJ-T vs. YK-T. We speculate that the plant hormone DAMs detected in this research might modulate the flavonoid and lignin biosynthesis downstream of phenylpropane metabolism in *Z. armatum* leaves. Interestingly, the correlation analysis revealed that three enzymes were significantly correlated with MeJA (*R* > 0.8, *p* < 0.01) (Com_522uneg, enriched in phenylalanine metabolism), namely, 4-coumaric acid coenzyme A ligase (4CL, Cluster-10777.11086), CHI (Cluster-10777.34174), and phenylalanine ammonia lyase (PAL, Cluster-10777.36576) (Figure 10E,F). This is consistent with our transcriptome and metabolome analyses, which further indicated that the phenylpropane metabolism pathway may be regulated by plant hormones, enabling Chinese pepper to resist rust. That is, *Z. armatum* could directly or indirectly regulate flavonoid biosynthesis genes through MeJA, thereby increasing the accumulation of various flavonoids and further enhancing its resistance to rust.

### 2.4. Differences in Flavonoid and Methyl Jasmonate Content in Z. armatum Leaves

Through the comprehensive analysis of the transcriptome and metabolome of the susceptible variety TJ and the resistant variety YK infected with rust, it was found that phenylpropane metabolism and the jasmonic acid pathway were the dominant factors controlling the rust resistance of *Z. armatum*. Therefore, the differences in flavonoid and MeJA contents in the YK and TJ leaves before and after inoculation were further analyzed. The contents of flavonoids and MeJA were significantly higher in YK-T than in YK-CK, and the content of MeJA was also significantly higher in TJ-T than in TJ-CK (Figure 11A), indicating that rust infection can induce the synthesis of flavonoids in resistant and susceptible varieties. At the same time, the susceptible variety TJ showed lower flavonoid and MeJA contents than YK, and MeJA biosynthesis was not significantly induced by rust infection (Figure 11B). The resistant variety YK has higher contents of flavonoids and MeJA than the susceptible variety TJ. With the significant enhancement of MeJA levels, flavonoid biosynthesis increases, and a positive relationship between the MeJA levels and flavonoid content was observed.

## 3. Discussion

Multiple factors, including climate change, changes in farming systems, and the widespread use of a low number of plant varieties, have led to a higher occurrence of plant diseases, which seriously threaten food security in China and globally [19]. Breeding and promoting the use of new disease-resistant varieties is an economically effective and environmentally friendly strategy to control crop diseases [20,21]. Therefore, resistance to many plant diseases, such as wheat rust [22], rice blast [23], and potato late blight [24], has been widely studied. Studying the disease resistance mechanisms of plants has an important guiding significance for the understanding, rational utilization, and precise modification of plants and improving their broad-spectrum resistance [25]. However, there are no reports on the resistance of *Z. armatum* to rust. Therefore, based on previous investigations, in this study, we determined the resistance of *Z. armatum* varieties to rust using artificial inoculation methods. The transcriptome and metabolome of susceptible varieties before and after inoculation were analyzed to study the molecular mechanisms underlying the resistance of different *Z. armatum* varieties to rust.

### 3.1. DEGs in Z. armatum Leaves Responsive to C. zanthoxyli Infection

Plant hormones are important plant defense response mediators in the interaction between plants and pathogens [26]. Different hormones mediate plant hormone signal transduction and have positive or negative regulatory effects on various nutritional pathogens [27], including activating the expression of defense genes in plants through plant hormone signaling, regulating the secondary metabolism, synthesizing plant antitoxin substances, and preventing the invasion and colonization of pathogens [28]. Most plant disease resistant substances can be synthesized through the phenylpropanoid metabolic pathway, such as flavonoids, terpenoids, alkaloids, etc., playing an important role in plant disease resistance and defense reactions [29]. In this study, DEGs rich in both the “plant hormone signal transduction” and “phenylpropanoid biosynthesis” pathways showed consistent results in the comparison of YK-CK vs. YK-T. This indicates that the metabolism of phenylpropanoid compounds may be regulated by plant hormones to activate the resistance of pepper to rust.

### 3.2. Metabolome Analysis Reveals the Potential Metabolites

The resistance of plants to pathogenic microorganisms largely depends on the synthesis of secondary metabolites [30]. Phenylpropane metabolism exhibits strong plasticity during plant development and in response to constantly changing environments. Phenylpropane metabolism is one of the most important secondary metabolic pathways in plants, producing over 8000 metabolites that play important roles in plant growth and development and plant–environment interactions [31]. Flavonoids are the most diverse branch of the phenylpropanoid metabolism pathway, consisting of approximately 6000 compounds [32,33,34,35,36], and are the most described secondary metabolites in the plant defense system. According to the change in the heterocyclic C-ring, flavonoids can be divided into chalcones, dragon ketones, flavonoids, isoflavonoids, dihydroflavonoids, leucoanthocyanidins, and anthocyanidins [35,36,37]. Flavonoids play important roles in plant resistance to pathogen invasions. In this study, the flavonoid content in *Z. armatum* leaves after the inoculation of the disease-resistant varieties was significantly increased, and the expression of the genes related to phenylpropanoid and flavonoid synthesis was also significantly upregulated. This is consistent with previous reports, indicating that the rapid accumulation of compounds in the phenylpropanoid synthesis pathway is related to disease resistance. For example, due to the increased accumulation of phenolic and flavonoid substances, the resistance of grapes and strawberries to *Botrytis cinerea* is enhanced [38]. Similarly, phenylpropanoid metabolism has been reported to enhance the resistance of the tobacco-to-tobacco mosaic virus (TMV) [39], the resistance of cherry to *Alternaria verticillata* [40], and the resistance of cotton to Verticillium wilt [41]. In cucumbers, many metabolites related to flavonoid biosynthesis are upregulated after infection with *Sphaerotheca fuliginea* in resistant varieties [42]. It has been reported that the resistance of alfalfa to fusarium wilt is closely related to the isoflavone pathway [43]. Yang et al. (2022) [44] found that the resistance of tobacco to Anthrax infection was enhanced by the overexpression of CoDFR. Three key genes (*CHS*, *CHI*, and *DFR*) for flavonoid biosynthesis were simultaneously expressed in flax by a transgenic method, which led to a significant increase in flavanones, flavonoids, flavonols, and anthocyanidin and improved the resistance of flax. These studies indicate that increasing the flavonoid content may play an important role in the formation of resistance. Pathogenic infection of resistant varieties can induce flavonoid biosynthesis in resistant varieties to enhance plant resistance. In combination with the transcriptomic and metabolomic analyses of *Z. armatum* and the determination of flavonoid contents, in this study, we found that the phenylpropane metabolism model of *Z. armatum* resistance to rust mainly belongs to the “induced” type, which is consistent with the previous studies of *Z. armatum* resistance to stem blight [45,46].

### 3.3. Integrated Analysis of the Transcriptome and Metabolome Reveals the Potential Disease Resistance Mechanism

Phenylpropane metabolism is regulated by a variety of regulatory pathways, such as transcriptional regulation, post-transcriptional regulation, post-translational regulation, and epigenetic regulation. In addition, plant hormones and biotic and abiotic stress affect phenylpropane metabolism [47]. In this study, it was found that a large amount of lipids and their metabolites were enriched in the metabolic pathway of disease-resistant varieties. An increasing body of evidence has shown that lipids are important regulators of plant defense [48,49,50]. Recent research also showed that lipids play important roles in inducing systemic acquired resistance [51,52]. Zhang et al. [53] found that exogenous MeJA mainly affected the biosynthesis of sesquiterpenes, triterpenes, and flavonoids in apple leaves. Zhan et al. [54] and other researchers found that the gene cluster DGC7 on rice chromosome 7 is directly regulated by the MeJA-mediated apparent regulator JMJ705 and can improve rice resistance to bacterial blight. According to Li et al., the addition of MeJA can enhance flavonoid accumulation in licorice cells [55,56,57,58]. In this study, through a further network analysis of the DEGs and DAMs of rust-resistant and -susceptible *Z. armatum* leaves, it was found that phenylpropane biosynthesis and the DAMs involved in flavonoid biosynthesis are regulated by DEGs related to the jasmonic acid pathway. For example, sesquiterpenes, triterpenes, and flavonoids are positively regulated by the MYC2 gene. Another interesting finding of this study is that 4CL, which is involved in the phenylpropane pathway, is significantly related to the DAMs (MeJA, Com_522uneg) of the jasmonic acid pathway (one of the top 10 DAMs upregulated by disease-resistant varieties). The resistance of *Z. armatum* to rust infection may be regulated by the co-expression of the DEGs and DAMs related to the biosynthesis of phenylpropane and flavonoids through the jasmonic acid pathway. Therefore, we speculate that the plant hormone lipid DAMs detected in this study, especially MeJA, may activate rust resistance in *Z. armatum* by regulating phenylpropane and flavonoid biosynthesis in *Z. armatum* leaves. The regulation of phenylpropane and flavonoid metabolism and MeJA levels in disease-resistant *Z. armatum* deserves further study, which will help to better understand the molecular mechanisms underlying MeJA-mediated disease resistance and defense.

To sum up, we identified rust-resistant and susceptible Chinese pepper varieties through artificial inoculation screening and then analyzed the physiological changes in resistant and susceptible varieties after infection with *C. zanthoxyli* based on transcriptomic, metabolomic, and physiological indicators. The results show that MeJA might activate rust resistance in *Z. armatum* by regulating the biosynthesis of phenylpropane and flavonoids in *Z. armatum* leaves. This finding was further supported by the expression of key genes related to phenylpropane metabolism and biosynthesis, as well as the differences in the flavonoid and MeJA contents of susceptible *Z. armatum*. The exploration of excellent genes for rust resistance in *Z. armatum* and the analysis of rust resistance mechanisms are key to achieving targeted breeding for rust resistance. The establishment of plant metabolite regulation mechanisms by tapping into the metabolome and transcriptome resources of *Z. armatum* is of great guiding significance for the disease resistance breeding of *Z. armatum* in the future and provides a scientific basis for further promoting the prevention and control of *Z. armatum* rust. However, the specific mechanism by which rust resistance in *Z. armatum* is regulated by MeJA levels, phenylpropane metabolism, and flavonoid metabolism is not completely clear, so this needs to be further clarified in the future.

## 4. Materials and Methods

### 4.1. Plant Material and Growth Conditions

Two *Z. armatum* varieties (TJ and YK) were employed in this study. Two-year-old seedlings of *Z. armatum* were kindly provided by a Chinese pepper orchard (located in Hongya County, Sichuan Province, China, 33°59′ N, 106°39′ E). Each variety used 20 Chinese pepper plants. Each seedling was planted in a plastic pot with an upper diameter of 30 cm, a lower diameter of 20 cm, and a depth of 20 cm containing approximately 8 L of nutrient soil and cultivated in an environmentally controlled greenhouse at Sichuan Agricultural University (Chengdu, China) at a temperature of 25 ± 2 °C, a relative humidity of 75%, and a photoperiod of 12 h of light/12 h of darkness with a light intensity of 2000 lx.

### 4.2. Inoculation of Chinese Pepper Leaves with C. zanthoxyli

The pathogenic fungus *C. zanthoxyli* isolated from *Z. armatum* leaves was provided by the Key Laboratory of Forest Protection of the province of Sichuan. The spores of *C. zanthoxyli* stored at 4 °C were scraped out using an inoculation loop, and the spore suspension was adjusted to 1 × 10^6^ spores/mL with sterilized water. Subsequently, 10 mL of *C. zanthoxyli* spore suspension was sprayed onto the leaves of 2-year-old seedlings of TJ and YK. After the suspension was absorbed by the Chinese pepper leaves, they were moved into a plastic basket and placed at room temperature (28 °C, humidity > 90%). The leaves of TJ and YK were collected at 0 and 12 dpi, quickly frozen in liquid nitrogen, and stored at −80 °C until further use. The experiment was carried out at least three times.

### 4.3. Transcriptome Sequencing and Data Analysis

Transcriptome analyses of the susceptible variety TJ and the resistant variety YK following *C. zanthoxyli* infection at 12 dpi were conducted, including controls (inoculation with sterilized water without *C. zanthoxyli*). The samples were taken from the diseased leaves (TJ-T and YK-T) and the control leaves (TJ-CK and YK-CK). Three biological replicates were collected for each treatment. Each replicate contained mixed sampled leaves from at least five sampled sites from different plants. All samples were immediately frozen in liquid nitrogen and stored at −80 °C.

Total RNA was extracted from Chinese pepper leaves using a Tiangen polysaccharide and polyphenol plant total RNA extraction kit (DP441). The Illumina NEBNext^®^ Ultra^TM^ RNA Library Prep Kit was used to construct the transcriptome library. Qualified libraries were sequenced on an Illumina NovaSeq 6000 platform. Raw counts were generated using Feature Counts v1.6.3 based on the reads mapped to the *Z. bungeanum* genome. Gene expression levels were determined using the FPKM method. DEGs were identified using DESeq2 v1.16.1 based on false discovery rate (FDR) < 0.05 and |log2(FC)| > 1. The R package clusterProfiler was used to perform the KEGG enrichment analysis. Pathways with FDR-corrected *p*-values of <0.05 were considered significantly enriched.

### 4.4. Metabolite Extraction and Detection

The leaves of TJ and YK following *C. zanthoxyli* infection at 12 dpi were further selected for metabolomic analysis of secondary metabolites. The sampling method was the same as for the transcriptome analysis experiments, and three biological replicates were set up for each treatment. The metabolome analysis was conducted by MetWare Biological Science and Technology Co., Ltd. (Wuhan, China) [59].

Samples were taken in the same manner and at the same sites as for the transcriptome analysis, with six sample replicates. Then, 100 mg of tissue was taken from each sample, ground in liquid nitrogen, and placed separately in EP tubes. Extraction was performed as follows: 500 µL of aqueous 80% methanol was added to a vortex suspension homogenate, followed by incubation on ice for 5 min and centrifugation at 15,000 rpm for 5 min at 4 °C. The supernatant was diluted with LC-MS grade water to a methanol concentration of 53%. Samples were centrifuged again at 15,000 rpm for 5 min at 4 °C, and the supernatant was injected into an LC-MS/MS system for analysis. UHPLC-MS/MS analysis was carried out using a Vanquish UHPLC system (Thermo Fisher, Braunschweig, Germany) and an Orbitrap Q Exactive TMHF-X mass spectrometer (Thermo Fisher, Germany) using default parameters at Novogene Ltd. (Beijing, China).

### 4.5. Data Analysis/Widely Targeted Metabolome Analysis

PCA and partial least squares discriminant analyses were performed using metaX, and VIP values were available for each metabolite. In the univariate analysis, statistical significance (*p*-value) and FC were calculated for metabolites based on *t*-tests. Quantitative analysis of metabolites was completed through multiple reaction monitoring [60,61]. The criteria for DEM screening are VIP > 1, *p* < 0.05, and FC ≥ 2 or FC ≤ 0.5. The volcano map was plotted using the R package ggplot2; the clustering heatmap was plotted using the R package Pheatmap; the metabolite data were normalized using z-scores; the Pearson correlation coefficient was determined using the R language cor; statistical significance was analyzed using cor.mtest in R; and the correlation map was plotted using the complot package in R. The KEGG database was used to study the metabolite function and metabolic pathways.

### 4.6. Correlation Analysis of Transcriptomic and Metabolomic Data

We selected three replicates from six independent metabolomic biological replicates for correlation analysis with transcriptome data. This correlation analysis was performed to determine the correlation between DEGs and DAMs. The Pearson correlation coefficient of the DEGs and DAMs was calculated using the cor function in R. DEGs and DAMs were simultaneously mapped to the KEGG database to identify their common pathways. The correlation coefficient between DEGs and DAMs was calculated, as well as the *p*-value. Genes and metabolites from common pathways were used to construct relevant network diagrams using the coefficient method. Network diagrams were visualized using Cytoscape version 3.7.1. Metabolic-transcriptional KEGG enrichment bubble maps were plotted for the co-enrichment pathways using the ggplot2 package in R.

### 4.7. qRT-PCR Analysis

To further confirm the expression levels of genes, qRT-PCR analyses were performed. We used cDNAs from sequenced samples as templates and randomly selected 10 genes with high differential multiplicity, including upregulated and downregulated DEGs, using β-actin as a reference gene [17]. qRT-PCR primers were designed using the coding sequences of 10 candidate genes using Premier 5.0 software (Appendix A). qRT-PCR analysis was performed on a CFX96^TM^ real-time PCR detection system (Bio-Rad, Hercules, CA, USA) using transScript^®^ Green One-Step qRT-PCR SuperMix (TransGen, Beijing, China) [62]. The primers used for qRT-PCR were designed using Premier 5 software (Premier Biosoft International, Palo Alto, CA, USA). All protocols were carried out according to the manufacturers’ instructions. qRT-PCR reaction mixtures contained 10 μL of mix (Beijing, China), 8 μL of ddH_2_O, 0.5 μL of F/R primer, and 1 μL of cDNA. The qRT-PCR program was as follows: initial denaturation at 94 °C for 20 s, followed by 38 cycles of 94 °C for 10 s and 60 °C for 20 s. Three replicates were performed. The expression levels of the target genes were calculated using the 2^−ΔΔCt^ method [63,64].

### 4.8. Quantitation of Total Flavonoid Content and Methyl Jasmonate in Z. armatum Leaves

The contents of the total flavonoids and MeJA in *Z. armatum* leaves under different treatments were further calculated. The sample groups were the same as those of the transcriptome analysis, namely, TJ-CK, TJ-T, YK-CK, and YK-T. The extraction of the total flavonoids was performed according to the previously reported method with slight modifications [65,66]. The MeJA content was determined using the enzyme-linked immunosorbent assay method [67].

### 4.9. Statistical Analysis

All data are presented as the mean ± standard deviation (SD) and were analyzed with SPSS Statistics 22.0. *p*-values of <0.05 were considered significant.

## Figures and Tables

**Figure 1 ijms-24-14761-f001:**
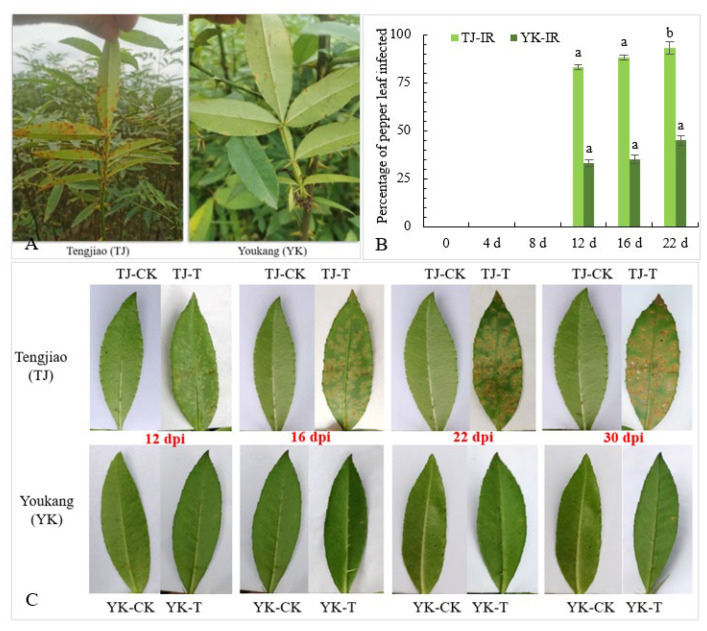
Differences in rust resistance between ‘Youkang’ (YK) and ‘Tengjiao’ (TJ). (**A**) The resistance of Chinese pepper cultivars YK and TJ to rust in the field. (**B**) The percentage of YK and TJ infected by *C. zanthoxyli*. IR, incidence rate. (**C**) The phenotypes of Chinese pepper cultivars YK and TJ inoculated with *C. zanthoxyli*. CK, control without inoculation; T, inoculation with *C. zanthoxyli*. Lesions were observed at 0, 6, 12, 16, 22, and 30 days post-inoculation (dpi). Different letters above the columns denote significant differences (*p* < 0.05).

**Figure 2 ijms-24-14761-f002:**
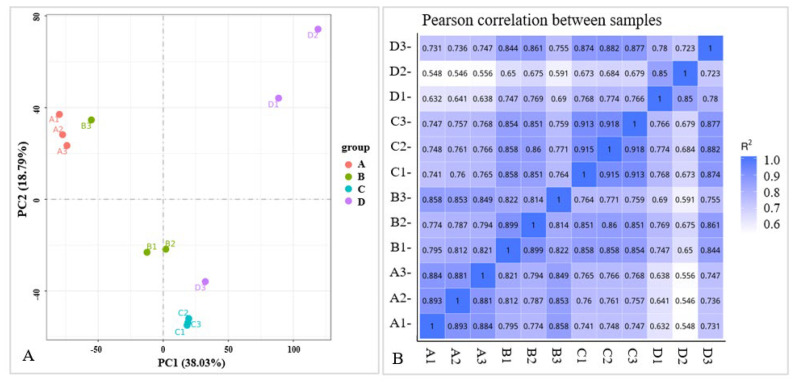
PCA and Pearson correlation analysis in different varieties of Chinese pepper. (**A**) PCA analysis in different varieties of Chinese pepper; (**B**) Pearson correlation analysis in different varieties of Chinese pepper. A1-3, YK-CK 1-3; B 1-3, YK-T 1-3; C1-3, TJ-CK 1-3; D1-3, TJ-T 1-3.

**Figure 3 ijms-24-14761-f003:**
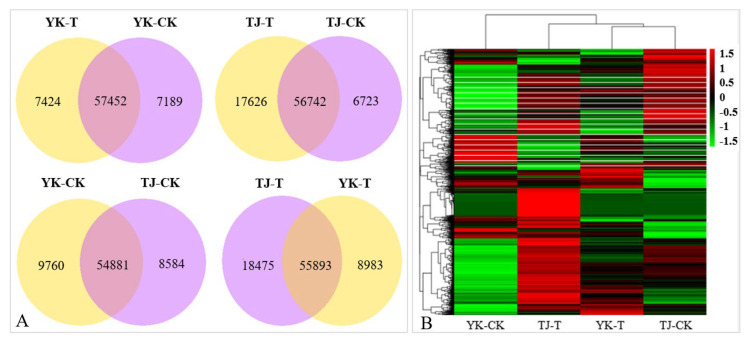
Overview of our transcriptome analysis of the *Zanthoxylum armatum* response to a *Coleosporium zanthoxyli* infection. (**A**) Venn graphs showing co-expression DEGs from the pairwise comparisons of four groups. (**B**) Heatmap visualization of DEGs. (**C**) Bar graph showing the numbers of up- and downregulated genes from pairwise comparisons. Blue is the total differentially expressed genes, red is upregulated, green is downregulated.

**Figure 4 ijms-24-14761-f004:**
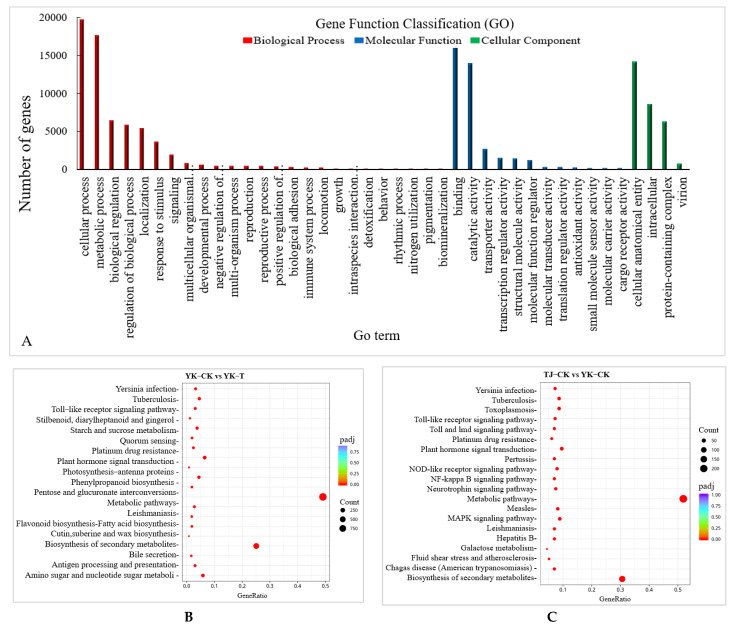
The GO annotation analysis and KEGG enrichment analysis of DAMs in different comparisons. (**A**) GO annotation analysis. (**B**) Significantly enriched KEGG pathways in the TJ-CK vs. TJ-T comparison. (**C**) Significantly enriched KEGG pathways in the YK-CK vs. YK-T comparison. (**D**) Significantly enriched KEGG pathways in the TJ-CK vs. YK-CK comparison. (**E**) Significantly enriched KEGG pathways in the TJ-T vs. YK-T comparison. Round dot indicate genes and triangle indicate metabolites.

**Figure 5 ijms-24-14761-f005:**
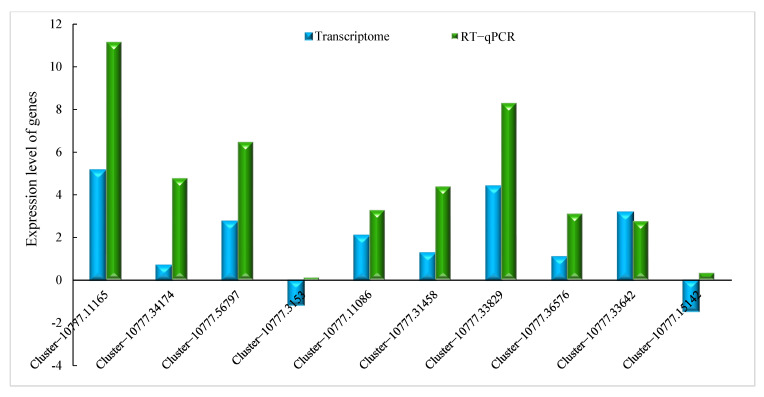
Comparison of the expression levels of genes in RNA-Seq and qRT–PCR analyses. Blue, expression level as determined by RNA–Seq; green, expression level as determined by qRT–PCR.

**Figure 6 ijms-24-14761-f006:**
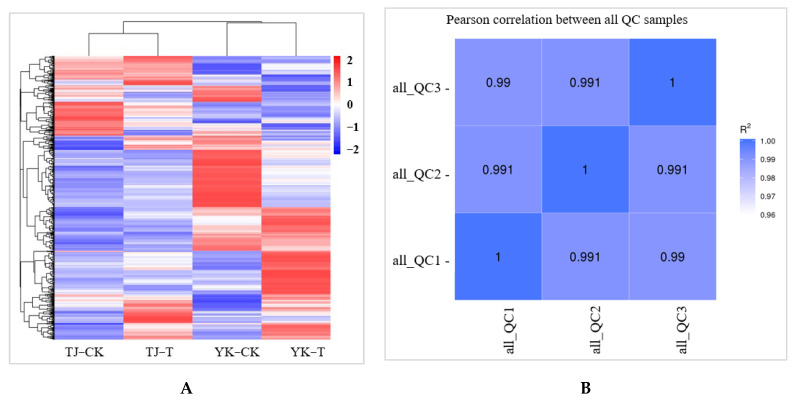
A chart of data quality control. (**A**) Heatmap of DAMs confirmed the significant differences in the metabolomes of TJ and YK; (**B**) The *R*^2^ value between the QC samples. QC, quality control.

**Figure 7 ijms-24-14761-f007:**
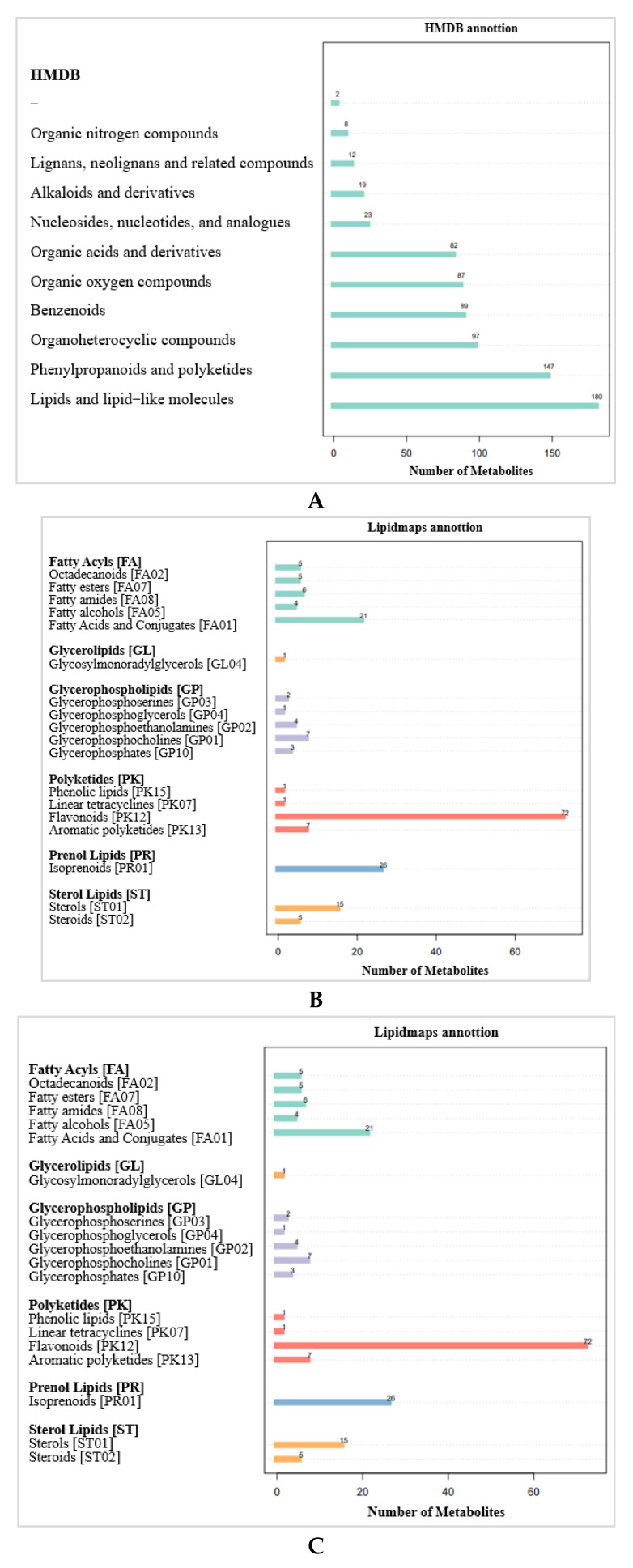
Annotation of DAMs. (**A**) HMDB annotation of DAMs. (**B**) KEGG pathway annotation of DAMs. (**C**) LIPIDMAPS annotation of DAMs.

**Figure 8 ijms-24-14761-f008:**
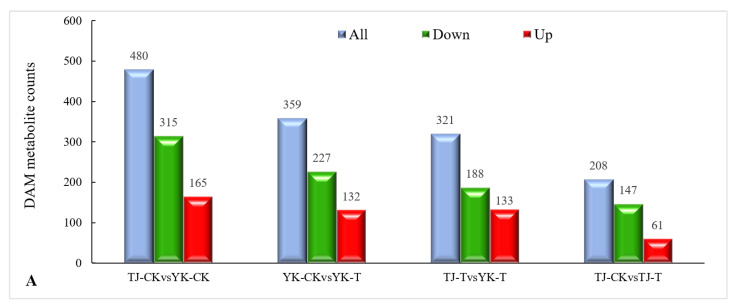
Overview of widely targeted metabolome analysis of *Z. armatum* in response to *C. zanthoxyli* infection. (**A**) Bar graph of up- and down-accumulated metabolites from pairwise comparisons. (**B**–**E**) KEGG pathway enrichment analysis of DAMs. (**B**) TJ-CK vs. TJ-T. (**C**) YK-CK vs. YK-T. (**D**) TJ-CK vs. YK-CK. (**E**) TJ-T vs. YK-T. The arrows highlight the significantly enriched metabolic pathways. Blue arrows indicate the pathways related to flavonoid metabolism. (**F**) Category and number of DAMs. The red star highlights phenylpropanoid and polyketide DAMs. Blue bars are the total differentially accumulated metabolites, red bars are up-accumulated, and green bars are down-accumulated.

**Figure 9 ijms-24-14761-f009:**
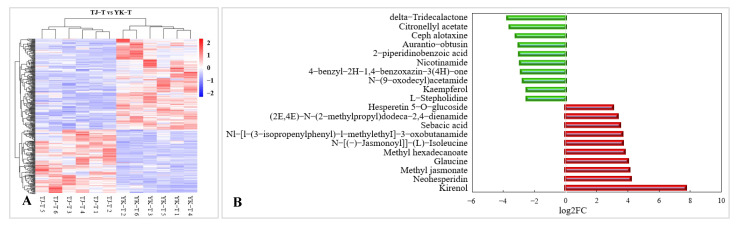
KEGG pathway enrichment analysis of DAMs and the top 10 DAMs. (**A**) KEGG pathway enrichment analysis of DAMs. (**B**) Top 10 up- and downregulated DAMs. Red bars indicate upregulated DAMs. Green bars indicate downregulated DAMs.

**Figure 10 ijms-24-14761-f010:**
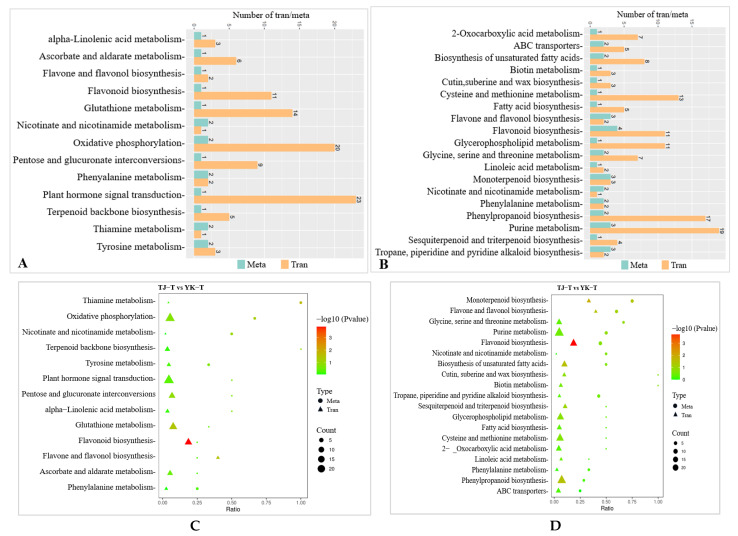
Network analysis of DEGs and DAMs of *Z. armatum* leaves following *C. zanthoxyli* infection. (**A**,**B**) Number of DEGs and DAMs in pos and neg. “Meta” and “Tran” are Metabolomic and Transcriptomic, respectively. “Pos” and “Neg” are positive and negative, respectively. (**C**,**D**) DEG and DAM enrichment in KEGG pathways in pos and neg. (**E**,**F**) Correlation network of DEGs and DAMs involved in phenylpropanoid biosynthesis in pos and neg. Blue rectangles indicate genes, and yellow rectangles indicate metabolites. Lines colored in red and blue represent positive and negative correlations, respectively, as determined based on a Pearson correlation coefficient of >0.8 or <−0.8, respectively.

**Figure 11 ijms-24-14761-f011:**
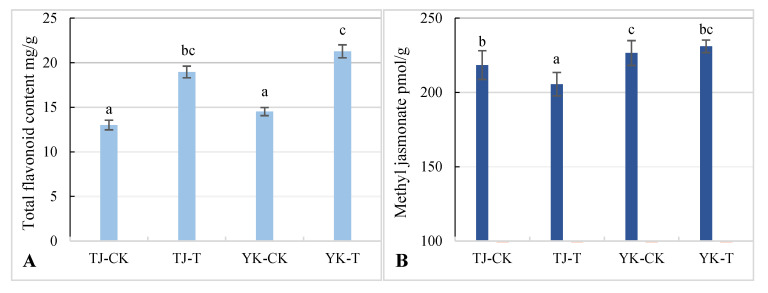
Analysis of flavone and MeJA contents in leaves of disease-resistant and susceptible *Z. armatum* varieties under different treatments. (**A**) Flavonoid content. (**B**) MeJA content. Different lower letters indicated significant differences at a level of *p* < 0.05.

**Table 1 ijms-24-14761-t001:** Sequencing data for the 12 samples.

Sample	Raw Reads	Clean Reads	Error Rate	Q20 (%)	Q30 (%)	GC Content (%)	Total Map
YK-CK1	21,440,315	20,913,184	0.03	97.27	92.57	44.83	31,488,252 (75.28%)
YK-CK2	23,545,476	22,713,650	0.03	97.54	93.12	44.24	33,838,906 (74.49%)
YK-CK3	22,641,971	22,074,057	0.03	97.32	92.62	44.5	32,675,810 (74.01%)
YK-T1	21,707,455	21,218,135	0.03	97.42	92.88	43.8	31,019,402 (73.10%)
YK-T2	21,024,152	20,494,901	0.03	97.26	92.52	44.2	30,387,514 (74.13%)
YK-T3	20,843,912	20,316,601	0.03	96.82	91.61	45.42	30,567,864 (75.23%)
TJ-CK1	20,666,347	20,205,862	0.03	96.88	91.72	43.78	29,411,490 (72.78%)
TJ-CK2	21,206,648	20,714,680	0.03	97.23	92.43	44.7	30,755,186 (74.24%)
TJ-CK3	22,574,829	22,034,849	0.03	97.1	92.2	45.06	32,950,058 (74.77%)
TJ-T1	22,062,869	21,543,733	0.03	97.09	92.19	43.99	31,528,218 (73.17%)
TJ-T2	22,507,430	21,693,488	0.03	97.21	92.43	43.74	32,443,130 (74.78%)
TJ-T3	22,944,166	22,082,650	0.03	97.47	92.96	43.84	33,163,596 (75.09%)

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
