# Peer review of "Integrated Transcriptome and Metabolome Analysis Reveals the Molecular Mechanism of Rust Resistance in Resistant (Youkang) and Susceptive (Tengjiao) Zanthoxylum armatum Cultivars"

_ijms, 2023, doi:10.3390/ijms241914761_

Round 1

Reviewer 1 Report

 The study presented in this manuscript is interesting and describes some new data about possible mechanism of rust resistance of Chinese pepper plants. In general, the study is well-designed, and obtained results have some value and prospects for those who work with this crop.

I do not have major comments to the study and its results, but have to say that both text and figures of the manuscript require some thorough examination to correct some uncertainties and inaccuracies, which complicate the perception and understanding of the presented data (see the list of minor comments). Note also that some figure parts have a poor resolution that should be also corrected. After such corrections, the manuscript can be published.

 TITLE

I would suppose some changes in the title to make it more correct:

Integrated Transcriptome and Metabolome Analysis Reveals the Molecular Mechanism of Rust Resistance IN RESISTANT (Youkang) and Susceptive (Tengjiao) Zanthoxylum armatum Cultivars

INTRO

 Line 75: it would be better to mention: which wild species (tomato?)

Line 76-77: please, re-phrase the sentence, since right now it looks like flavonoid biosynthesis regulates resistance to dry rot by combining transcriptomic and metabolomic analyses, i.e., this process uses these two analyses to regulate resistance.

RESULTS

 Fig. 1B: what is IR in TJ-IR and YK-IR. Please, explain in the figure caption. Also, B and C parts of the figure do not correspond to the corresponding parts of the figure caption. Please, check.

Fig. 2A: please, add the details for A, B, C, and D variants to the figure caption, since figures should be self-explanatory.

Fig. 3A: it is unclear, where the diagram shows upregulated genes and where - downregulated. Please, add explanation (designations) to the diagram.

Line 194: Fig. 4A, B, D does not contain the groups “phenylpropanoid biosynthesis” and “MAPK signaling pathway”. Actually, the first group presents in Fig. 4C, while the second presents in Fig. 4B,D. Please, check the text and correct, if necessary.

Line 240: what is QC? This is the first time this abbreviation appeared in the text.

Fig. 6. Please, check and correct the figure caption since it copies that of Fig. 5.

Fig. 7: poor image quality, especially for the C part. The same is for Fig. 8 (excepting A- and F-parts).

Subsection 2.3.3: please, check the end of rows since some words are broken (c onducted, a nd, show n, etc.)

Line 263 and 267: Fig. 6 or Fig. 8? Please, check.

General comments for diagrams reflecting all, down-, and up-regulated genes, DAMs, etc. It would be good to use a unified color designation to avoid misunderstanding. For example, Fig. 8F shows upregulated DAMs in green, while down-regulated DAMs in red (according to a common practice). However, DAM metabolite counts shown in Fig. 8A have the opposite color marking (green for down and red for up).

Line 289-290: which arrows?? No any arrows in Figure 8. Please, check. What is flenriche metabolism?

Line 291: There is no red star on the Figure. Please, check.

Line 338-339: “might modulate fl”? “d and lignin biosynthesis”? Please, check the text and correct, if necessary.

Fig. 10A,B: what are meta and tran? Please, give explanation in the figure caption. What do the A and B parts show (no explanation in the figure caption and on diagrams)? The same question is for C and D, E and F parts. What do “pos” and “neg” in the figure caption mean? Please, give a more detailed figure caption or add some explanations on diagrams. VERY poor quality of the E and F parts: it is impossible to read anything.

Fig. 11 is lost.

Line 365: please, check the text: “was also significantly higher in TJ-T than in TJ-T”

MATERIALS AND METHODS

 Subsection 4.1: how many plants were used for each variety?

Subsection 4.2,

-          Line 487: what buffer (or water?) was used to adjust spore suspension?

-          Line 496: what is PDA? Please, add the full name.

English Language is acceptable, i.e., only some minor editing is required.

Reviewer 2 Report

No quantitative information is present in the abstract. Add a few information related to the major DEGs and DAMS details.

The introduction should clearly explain the scope, importance, and incentive of the work. Add recent relevant studies and clearly write the main objectives at the end of the introduction.

How is possible to be sure that these primers amplify only a specific gene and not the others of the same family? Authors must provide full information about all used primers. These informations have to include: (A) Names of primers; (B) Sequence accession numbers from the database, for example, NCBI (Like: AB12345)  or another database for each gene studied; (C) Sequences of both F and R primers; (D) Amplicon sizes for each used qPCR product; (E) Reference gene GAPDH must be also included in this list with all required information; (F) If any of the presented information about primers and qPCR in mung-bean (including reference gene GAPDH) was published, please provide references. This information has to be present in the Supplementary file with full details.

All the figures need to improve. Give high-quality images for figures. Figures 5 and 12 add the statistical information.

I suggest selecting the gene transcripts based on the transcriptome and metabolome association/integration analysis and validating them by qRT-PCR analysis.

The discussion needs subsequent improvement, divide it into subsections and discuss more detail with recent results. 

Minor editing of the english language required

Reviewer 3 Report

This study showed that integrated transcriptome and metabolome analysis revealed the molecular mechanism of rust cultivars. The study design is accptable. The study contains some valuable results that can be considered for publication. 

Suggestion:

Figure 2: Figures are small. difficult to read the numbers.

Figure 3: Give in full the latin names in the title.

Figure 4: B, C, D, E: difficult to read (too small letters) 

Figure 7: Again too small letters in the figure. Give abbreviations in full in the title. 

Figure 8: B,C - again small letters.

Figure 10: Very-very small letters. Unreadable for E and F.

Round 2

Reviewer 2 Report

I recommend the manuscript for publication...